# Rising Concern About the Carcinogenetic Role of Micro-Nanoplastics

**DOI:** 10.3390/ijms26010215

**Published:** 2024-12-30

**Authors:** Lorenzo Ruggieri, Ottavia Amato, Cristina Marrazzo, Manuela Nebuloni, Davide Dalu, Maria Silvia Cona, Anna Gambaro, Eliana Rulli, Nicla La Verde

**Affiliations:** 1Department of Oncology, Luigi Sacco University Hospital, ASST Fatebenefratelli Sacco, 20157 Milan, Italy; ruggieri.lorenzo@asst-fbf-sacco.it (L.R.); amato.ottavia@asst-fbf-sacco.it (O.A.); marrazzo.cristina@asst-fbf-sacco.it (C.M.); dalu.davide@asst-fbf-sacco.it (D.D.); cona.silvia@asst-fbf-sacco.it (M.S.C.); gambaro.anna@asst-fbf-sacco.it (A.G.); 2Pathology Unit, Luigi University Hospital, ASST Fatebenefratelli Sacco, 20157 Milan, Italy; nebuloni.manuela@asst-fbf-sacco.it; 3Methodology for Clinical Research Laboratory, Clinical Oncology Department, Istituto di Ricerche Farmacologiche Mario Negri IRCCS, 20156 Milan, Italy; eliana.rulli@marionegri.it

**Keywords:** microplastics, nanoplastics, ecotoxicology, tumor, cancer, carcinogenic, carcinogenicity, cancerogenesis, cancerogenicity

## Abstract

In recent years, awareness regarding micro-nanoplastics’ (MNPs) potential effects on human health has progressively increased. Despite a large body of evidence regarding the origin and distribution of MNPs in the environment, their impact on human health remains to be determined. In this context, there is a major need to address their potential carcinogenic risks, since MNPs could hypothetically mediate direct and indirect carcinogenic effects, the latter mediated by particle-linked chemical carcinogens. Currently, evidence in this field is scarce and heterogeneous, but the reported increased incidence of malignant tumors among younger populations, together with the ubiquitous environmental abundance of MNPs, are rising a global concern regarding the possible role of MNPs in the development and progression of cancer. In this review, we provide an overview of the currently available evidence in eco-toxicology, as well as methods for the identification and characterization of environmental MNP particulates and their health-associated risks, with a focus on cancer. In addition, we suggest possible routes for future research in order to unravel the carcinogenetic potential of MNP exposure and to understand prognostic and preventive implications of intratumoral MNPs.

## 1. Introduction

Pollution is a primary environmental cause of disease of the contemporary era [1]. In recent years, starting from concerns about plastic sea pollution, outstanding research efforts have provided evidence regarding the widespread presence of micro-nanoplastic (MNP) litter in different environments worldwide [2,3,4,5]. Historically, soon after the laboratory development of first-in-class plastic polymers in the early 1900s, the rise of commercial plastics exponentially increased during the 20th century, ever since the large-scale extraction and utilization of oil-based products. Indeed, plastic is a class of multiple synthetic, carbon-based polymers synthetized through the manipulation of oil components. Plastic revolutionized modern industry, progressively replacing several natural materials short in supply, witnessing continuous expansion since the 1960s and a 20-fold production increase in a 50-year period [6]. On the other hand, belated maturation of ecological policies lead to the employment of inappropriate waste methods, causing severe environmental consequences, as a great amount of mismanaged plastic accumulated in multiple ecosystems. Over time, environmental weathering processes such as ultraviolet radiation (UV photo-oxidation), wind and sea waves caused the progressive fragmentation of plastic items (macroplastics) into microscopic particles [7]. The formation of this particulate, defined as MNP debris, now constitutes an environmental litter that permeates the terrestrial and marine environments and that has infiltrated multiple ecosystems. Furthermore, MNP debris can also recirculate in the water cycle through atmospheric suspension, aggravating its diffusion and interaction with living beings and humans and fostering continuous ultra-fragmentation. Thus, since their physical resistance, MNP particles are persistently recirculating across ecological systems [8] (see Figure 1).

MNP debris is composed of microplastics (MPs) and nanoplastics (NPs). MPs are defined as plastic particles ranging between 1 millimeter and 1 micrometer, while NPs range between 1 micrometer and 1 nanometer [9].

Since the amount of mismanaged macroplastics exponentially increased during the last decade [10], in 2019, the Scientific Advice Mechanism Group provided the European Commission a scientific opinion on MNPs. They concluded that, while MNP pollution does not currently represent a widespread risk, stable emissions at the present rate will potentially cause serious damage within a century, warranting the introduction of preventive measures [11].

To date, preclinical evidence of a possible role of MNPs in the etiology of multiple human diseases is growing [12]. Regarding cancer, epidemiological evidence is scarce, and MNPs are not counted as potential carcinogens by the International Agency for Research on Cancer (agents classified by the IARC Monographs, Volumes 1–136) [13]. On the other hand, multiple preclinical and clinical studies have recently highlighted a potential link between MNP exposure and cancer development. In this narrative review, we will offer an overall prospect of the main good-quality evidence available, offering general considerations about eco-toxicology, risks for human health and potential carcinogenic effects of MNPs, providing a future perspective on cancer research from a medical oncologist point-of-view.

## 2. Definition and Eco-Toxicology of MNPs

### 2.1. Definitions and General Considerations

MNP debris is a highly complex entity, originating from multiple sources. In general, MNPs are classified as primary or secondary. Primary MNPs, such as plastic micro-beads added to cosmetics, are microscopic plastics produced for a specific purpose. Conversely, secondary MNPs originate from mismanaged environmental macroplastics [14]. Secondary MNPs represent the greatest bulk of environmental MNPs, originating through the abrasion or degradation of multiple plastic-based materials, such as synthetic cloth textiles, car tires, plastic coatings, paints and insulating materials. These contribute to the formation of city dust and marine MNP particulates, the latter enlarged by the abrasion of marine vessels and fishing gear [6].

Derived from a wide spectrum of different polymers, MNPs have extremely variable chemical compositions. The main components of commercial macroplastics that form MNPs are polyethylene, polyamide, polystyrene, polypropylene, polyvinyl chloride, polyurethane, polyacrylonitrile and other minor plastic polymers [15]. Since the availability of reliable methods to detect and track MNPs is currently limited, estimations of the quantitative proportional contribution by different sources remains controversial [6]. Similarly, while MNPs are known to recirculate in the environment, their whole “life-cycle” has to be established with further research, further limiting estimations about their origin and distribution [6].

In this context, microscopic characterization of environmental MNPs highlighted that a peripheral, biochemical, layered structure called eco-corona develops around MNP particles [16]. These layers develop when MNPs encounter environmental natural organic matter, generally organic polymeric substances such as DNA, proteins, carbohydrates and humus substances, through chemical interactions as hydrogen bonds, van der Waals forces and hydrophobic interactions. Thus, the eco-corona can further enrich the spectrum of bio-physico-chemical interactions between MNPs, environment and living organisms [17,18,19,20], adding another level of complexity in their ecological interplay.

However, current estimations suggest that the increasing prevalence of plastic use in multiple aspects of daily human life caused a progressive spread of MNPs in the environment. Multiple sources of human exposure have been described, as via the inhalation of particulate matter and other airborne particles, via exposure to beauty and human care products and via the consumption of contaminated foods and beverages [6,12,14,21,22,23]. Indeed, MNPs have been isolated across most marine environments—both in the abyss and on the coastline, accumulating on the surface of sandy beaches—and in freshwaters, the latter with proportional concentrations in correlation with the intensity of human activities in the area [24,25,26,27,28,29,30,31]. MNPs have also been detected in soils across most continents, both in agricultural fields and in areas not subject to human activities, probably coming mainly from fertilizers and sewage debris. Interpretation of these data remains biased by technical limitations in MNP isolation methods [3,32,33,34,35,36]. Moreover, MNPs are present both in indoor and outdoor air, with two-fold higher concentrations reported for the former, primarily in urban environments [37,38,39,40,41,42]. Finally, wastewaters of urban areas are largely contaminated, derived from domestic and industrial sources [43,44,45,46]. Considering the widespread contamination of aquatic ecosystems, humans are exposed through water and food, as bottled and tap drinking water, beer, milk, honey, rice, sea salt, seafood and fish. Moreover, crop contamination from soils and fertilizers has been described [36,47,48].

Nowadays, the quality of the available evidence lacks in comprehensive geographical and local characterization. Only a minority of studies on MNPs’ diffusion in freshwater have been performed in Asia, while South America and Africa are overall marginalized [13]. In this regard, the abovementioned scientific opinion provided to the European Commission in 2019 and the more recent report on dietary sources of MNPs issued by the World Health Organization in 2022 emphasized that evidence is limited to date. Moreover, criticism has been expressed about the quality of most of the studies, concluding that human MNP exposure from environmental sources and its related risks remain unknown [14,47]. Moreover, considering the nature of eco-coronate MNPs, these structures account for additional possible biological hazards of MNP debris, such as the formation of micro-organism colonies on the surface, entrapment of toxic chemicals and release of toxic additives [14,49,50]. Thus, predicting MNPs’ effects in different environmental and biologic systems, as well as providing a general risk assessment, is extremely complex according to the current knowledge [14].

### 2.2. Routes of Human Exposure

Humans are exposed to MNPs primarily through ingestion, inhalation and dermal absorption [51]. As described above, MNPs can be ingested through drinking water and food. When ingested, these particles accumulate in the gastrointestinal tract lumen, where they adhere to the mucous layer. Large particles interact with M-cells, while particles smaller than 50 µm can be absorbed through enterocytes. Moreover, small particles of various size can cross intercellular spaces [12,52,53]. Penetrating the mucosal layer, MNPs can reach the bloodstream and lymphatic systems [54].

Airborne MNPs can be inhaled as suspended particles, especially in urban and industrial environments. Conversely, rural sources have rarely been described [55,56,57]. Particles below 2.5–3 µm have a higher probability of reaching alveolar sacs, depositing in lung tissue [58,59,60] or crossing the alveolar–capillary barrier, entering the bloodstream. Alternatively, MNPs are engulfed by alveolar macrophages [61]. Moreover, airborne MNPs can be absorbed through the olfactory bulb, potentially reaching the central nervous system [62].

Airborne MNPs can also be absorbed through dermal contact. Indeed, in vitro evidence of active micropinocytosis of keratin-coated MNPs by keratinocytes suggests that skin could represent a locus minoris resistentiae through which MNP particles can penetrate the body [63].

## 3. Analytical Methods of MNPs in Biological Studies

In recent years, several efforts have been focused on improving methods for MNP detection in different media. In 2023, the International Organization for Standardization (ISO) published the international standard ISO 4484-2:2023, establishing a qualitative–quantitative analytical evaluation of microplastics in the textile sector, with standards for determining their particle number, morphology, dimensional distribution and chemical nature in media such as textile process wastewater, clothes-washing water, textile process air emissions and textile process solid waste [64]. However, at present, methods of detection, identification and characterization of MNPs in biological studies are heterogeneous, and analytical processes are far from being standardized. Differences in MNP size definitions and limitations in analytical power hamper the reproducibility and cross-comparison of results, and further research is needed to refine methods and define standardized protocols [65]. Currently, different approaches could be informative of multiple characteristics of MNPs. Cell models could describe cellular uptake and intracellular localization. Animal models dissect bioaccumulation tendencies and systemic and local effects. In vivo imaging models track MNP translocations in blood circulation and tissue penetration. Finally, ex vivo studies can generate real-life information on MNP distribution and in-vivo effect estimation. Here, we provide an overview of the analytical methods used to characterize the biology of MNPs.

### 3.1. Surrogate Model Studies

MNPs can primarily recirculate through the bloodstream and lymphatic system, potentially reaching various organs and tissues. To date, human body circulation, distribution and tissue accumulation are estimated mainly using surrogate in vivo animal models, in vitro cell cultures and computational models. In vivo animal studies are used to evaluate how MNPs enter the body through ingestion or inhalation. The particles can be tracked using fluorescent dyes, allowing researchers to observe their distribution in various organs like the lungs, liver and kidneys [66]. On the other hand, cell cultures give a significant opportunity to examine direct interaction of MNPs and human cells. These studies can elucidate different mechanisms of active cell uptake and intracellular processes of MNPs [67]. In recent years, computational models have gained attention for their capability to predict biological and chemical behaviors. Simulations of MNP distribution throughout the body offer reliable predictions based on available data. These models could inspire the design of more informative future studies that will help to dissect MNP recirculation and accumulation in human tissues.

### 3.2. In Vivo Imaging Techniques

Magnetic resonance imaging (MRI) and positron emission tomography (PET) are used to monitor the distribution of traceable MNPs in the body in real-time. PET’s ability to provide real-time imaging enables one to track MNPs’ passage through the bloodstream, absorption by organs and tissue accumulation. This helps in distinguishing between acute (short-term) and chronic (long-term) exposures, assessing timings of MNP persistence in the body and determining sites of primary accumulation and main routes of elimination [68]. Another recent method that aims to dissect MNP interactions with cells and tissues is fluorescent labeling, a technique that can track real-time MNP movements [66].

### 3.3. Ex Vivo Studies

Techniques like electron microscopy, fluorescence, Raman spectroscopy and Fourier-transform infrared (FTIR) spectroscopy are employed to visualize and identify MNPs in ex vivo biological tissues, enabling the direct observation of the particles within organs [69,70,71,72]. FTIR spectroscopy and Raman spectroscopy are vibrational spectroscopy methods widely used to identify MNPs in human tissues [73,74,75]. They can assess polymer type, size, shape and distribution. FTIR spectroscopy is particularly useful in analyzing particles larger than 10 µm, while Raman spectroscopy can be used for even smaller particles, down to 1.2 µm. Electron microscopy, as scanning electron microscopy (SEM) and transmission electron microscopy (TEM), is employed to visualize and measure nanoplastics in ex vivo samples [72]. These methods provide detailed images of particles, allowing researchers to assess their morphology and distribution at the cellular level.

## 4. Human Health-Risk of MNP Exposure

Evidence is accumulating regarding MNP-related health risks, suggesting multiple molecular mechanisms as drivers, mainly through induced persistent chronic inflammation, oxidative stress and the disruption of other physiological processes. MNPs can induce a variety of cytotoxic effects, mainly through reactive oxygen species (ROS) generation, which can lead to oxidative stress, inflammation and cellular damage [76]. ROS play a significant role in the disruption of cellular homeostasis, contributing to inflammatory responses by activating multiple molecular pathways [77]. The excessive production of ROS causes oxidative damage to lipids, proteins and DNA [78]. Studies in both in vitro and in vivo models have shown that exposure to MNPs significantly increases ROS levels, leading to cellular dysfunction and apoptosis [79,80,81]. Indeed, the imbalance between ROS production and the antioxidant defense system can activate various signaling pathways associated with inflammation, such as the nuclear factor-kappa B (NF-κB) pathway [82]. This pathway is known to regulate the expression of pro-inflammatory cytokines and can be activated in response to cellular stress caused by MNPs [83]. The prolonged activation of inflammatory pathways may lead to chronic inflammation, which has been linked to a range of pathological conditions, including fibrosis and tissue damage [84]. In addition, MNPs can perturb mitochondrial metabolism until apoptosis occurs [85]. Furthermore, research has shown that MNPs could interfere with autophagy, causing potentially harmful cell damage [86,87,88].

### 4.1. Digestive Health

Ingested MNPs tend to be processed by stomach and gut secretions, going through progressive fragmentation [89]. The systemic toxicity of ingested MNPs is mainly related to the disruption of the gut microbiome, leading to the perturbation of gut–organ axes and of systemic immunological modulation [90]. Locally, MNP accumulation in mucosal cells and the lymphoid-associated structures of digestive tissue can induce chronic inflammation and direct cell death [91].

### 4.2. Respiratory Health

Evidence regarding MNPs’ lung toxicity in humans mainly derives from occupational exposure studies. Workers of the plastic industry exposed to plastic flocks have a reportedly higher incidence of pulmonary fibrosis and pneumoconiosis [92,93,94,95]. Indeed, preclinical models have shown that persistent alveolar macrophage activation and alveolar cell damage are the main underlying mechanisms involved in the pathogenesis of chronic lung damage [96].

### 4.3. Cardiovascular Health

Recent studies have linked MNPs to cardiovascular diseases. Rats exposed to polystyrene MNPs at incremental environmental concentrations developed proportional grades of cardiac fibrosis through oxidative stress mechanisms [97]. In another mouse model study, inhaled NPs accumulated preferentially in inflamed vascular lesions [98]. In this regard, a recent pivotal prospective analysis of resected atheromas from carotid endarterectomies showed that the presence of MNPs in macrophages or in the plaques is a negative predictive factor for secondary cardiovascular events and mortality. This was the first study to indirectly prove that MNPs could induce or foster inflammation in humans [72].

### 4.4. Endocrine Disruption

Rather than having a direct endocrine disrupting effect, MNPs have been reported to be a potential vectors of other substances known to have the potential to alter multiple endocrine axes, including phthalates, bisphenols, tributyltin and per- and polyfluoroalkyl derivatives [99,100,101,102]. However, data from mouse models suggest potential direct endocrine disturbances of reproductive functions by polystyrene nanoplastics, possibly mediated by DNA damage leading to alterations in sperm morphology and viability [103]. In addition, there are reports suggesting that MNPs might affect the reproductive function not only by inducing oxidative and inflammatory damage to gonadal tissues, but also via the down-regulation of genes involved in steroidogenesis [104].

## 5. MNPs’ Role in Cancer Development and Progression

Available evidence on the potential role of MNPs in cancer initiation and progression emerges mainly from preclinical studies of in vitro models. These studies have focused on the capability of MNPs to elicit a number of different biological effects potentially linked to carcinogenesis and are summarized in Table 1. Key biological processes that influence the induction and promotion of carcinogenesis include tumor-promoting inflammation, oxidative stress, alterations in cellular metabolism, genotoxicity, cytotoxicity, induction of invasive cellular phenotypes and changes in the local microbiome [105].

### 5.1. Tumor-Promoting Inflammation

A body of studies assessed MNPs’ capabilities of causing a sustained inflammatory response, as a surrogate of their carcinogenicity potential, according to the hypothetic carcinogenetic capacity of chronic inflammation in mammals. Human cancer cell lines significantly increased their production of inflammatory cytokines when exposed to polystyrene MNPs [106,107,108,109]. In vivo evidence in mammalian models confirmed the capacity of polystyrene MNPs to induce variegated mucosal inflammatory response [110,111,112,113,114]. In addition, abundant evidence on mouse models showed that oxidative stress induced by MNPs causes endoplasmic reticulum stress and other intracellular injuries, ending up in fatal processes as apoptosis, ferroptosis, pyroptosis and necroptosis in potentially all organs [115,116,117,118,119,120,121,122,123,124], further enhancing tissue inflammation.

### 5.2. Cancerogenesis Initiation

Different types of alterations in determined cellular processes could initiate the multi-step process of carcinogenesis, generally through the loss of apoptotic capacity and anchorage-dependent growth inhibition. Regarding the latter, previous studies demonstrated that the exposure of human cancer cell lines to MNPs can lead to anchorage-independent outgrowth [125,126]. Immune surveillance is another fundamental mechanism in the prevention of cancerogenesis initiation in mammals. Regarding this, MNPs have shown the potential to polarize immune responses in pregnant mice towards immune tolerance, through a decrease in NK cells and M2 polarization of macrophages [127]. In a similar manner, when Type-1-diabetic mice were exposed to MNPs, apoptosis of CD4+ T-lymphocyte increased in favor of an expansion of Tregs and complete reversion of autoimmune response [128]. In addition, MNPs showed genotoxic potential in human blood cells and mammalian cell models in vitro [129,130,131,132].

**Table 1 ijms-26-00215-t001:** Summary of studies evaluating the carcinogenic potential of MNPs exposure in multiple in vitro and murine in vivo models.

Study	Carcinogenic Potential	Biological Process	Experimental Model	Type of MNP
Forte et al., 2016 [106]	Tumor-promoting inflammation and cancerogenesis promotion	Increased production of inflammatory cytokines; dysregulated cell proliferation	Human gastric adenocarcinoma cell lines in vitro	44 nm and 100 nm polystyrene nanoparticles
Xu et al., 2022 [107]	Tumor-promoting inflammation	Increased production of inflammatory cytokines	Human colon adenocarcinoma cells in vitro	Polyurethane microplastic
Shi et al., 2024 [108]	Tumor-promoting inflammation	Increased production of inflammatory cytokines	Mice in vivo	100 nm polystyrene nanoparticles
Cheng et al., 2022 [109]	Tumor-promoting inflammation	Increased production of inflammatory cytokines	Liver organoids in vitro	1 μm polystyrene microbeads
Lu et al., 2021 [110]	Tumor-promoting inflammation	Mucosal inflammatory response	Mice in vivo	Cospheric microspheres
Choi et al., 2021 [111]	Tumor-promoting inflammation	Mucosal inflammatory response	Mice in vivo	0.5 μm polystyrene microplastics
Li et al., 2020 [112]	Tumor-promoting inflammation	Mucosal inflammatory response	Mice in vivo	Polyethylene microplastics
Li et al., 2022 [113]	Tumor-promoting inflammation	Mucosal inflammatory response	Mice in vivo	Polystyrene nanoparticles
Deng et al., 2017 [114]	Tumor-promoting inflammation	Mucosal inflammatory response	Mice in vivo	5 μm and 20 μm polystyrene microplastics
Barguilla et al., 2022 [125]	Cancerogenesis initiation and promotion	Anchorage-independent outgrowth; dysregulated cell proliferation	Mouse cell lines in vitro	Polystyrene nanoplastics
Barguilla et al., 2022 [126]	Cancerogenesis initiation and promotion	Anchorage-independent outgrowth; dysregulated cell proliferation	Mouse cell lines in vitro	Polystyrene nanoplastics +/− arsenic
Hu et al., 2021 [127]	Cancerogenesis initiation	Increased immune tolerance	Mice in vivo	Polystyrene microplastics
Liu et al., 2020 [128]	Cancerogenesis initiation	Increased immune tolerance	Mice in vivo	Poly[lactic-co-(glycolic acid) microparticles
Gopinath et al., 2019 [129]	Cancerogenesis initiation	Genotoxicity	Human blood cells in vitro	100 nm polystyrene nanoparticles
Ballesteros et al., 2020 [130]	Cancerogenesis initiation	Genotoxicity	Human blood cells in vitro	Polystyrene nanoplastics
Poma et al., 2019 [131]	Cancerogenesis initiation	Genotoxicity	Human cell lines in vitro	Polystyrene nanoplastics
Rubio et al., 2020 [132]	Cancerogenesis initiation	Genotoxicity	Human cell lines in vitro	Polystyrene nanoplastics
Xu et al., 2019 [133]	Cancerogenesis promotion	Dysregulated cell proliferation	Human cell lines in vitro	25 nm and 70 nm polystyrene nanoparticles
Halimu et al., 2022 [134]	Cancerogenesis promotion	Epithelial-mesenchymal transition	Human cell lines in vitro	Polystyrene nanoplastics
Kim et al., 2022 [135]	Cancerogenesis promotion	Epithelial-mesenchymal transition	Mice in vivo	Polystyrene microplastics
Traversa et al., 2024 [136]	Cancerogenesis promotion	Epithelial-mesenchymal transition	Human cell lines in vitro	Polyethylene MNPs
Shao et al., 2024 [137]	Cancerogenesis promotion	Epithelial-mesenchymal transition	Mice in vivo	100 nm nanoplastics

### 5.3. Cancerogenesis Promotion

Once pre-cancerous cells gain immortalization and escape the immune system, they need to acquire determined genetic mutations in order to progress in the carcinogenetic process. This phase, called “promotion”, involves a series of alterations in cell proliferation and invasiveness that ultimately result in the development of invasive disease. Different cellular effects of MNP exposure are related to the dysregulation of cell proliferation [106,125,126,133], potentially fostering the development of cancer. Moreover, in vitro models show that MNP exposure can induce epithelial–mesenchymal transition in human cell lines [134,135,136,137], a process that enhances invasiveness and migration capability of cancer cells.

### 5.4. MNPs’ Presence in Cancer Tissue

To date, the available studies show that MNPs in human cancer tissues are generally more abundant than in non-cancerous tissues. In an historical study, MNPs were more commonly detected in lung cancer than in normal lung tissue [138]. In a more recent case–control study, MNPs were significantly more concentrated in colorectal cancer tissue than in non-tumoral tissue [139]. In another recent study, the abundance of MNPs in tumor and para-tumoral tissues of human prostate cancer were compared, showing that particles were significantly more concentrated in cancerous tissue [140]. Underlying mechanisms should be unraveled, and realistic hypotheses could be hard to propose. Since MNPs can be inherited by cell division and actively absorbed by cancer cells, the higher concentration in tumor tissue could be related to a higher absorption rate of circulating and tissue MNPs. Common alterations in intratumoral circulation and the microenvironment could be responsible for an increased diffusion of MNPs within the tumor, together with a possible fostered pinocytosis of cancer cells. In this scenario, the abundance of MNPs in the tumor tissue is merely representative of their specific bioaccumulation tendencies and overall individual exposure. On the other hand, MNPs’ capability to foster carcinogenic events could influence clonal evolution, favoring the proliferation of cells with more effective pinocytosis processes. Extensive characterizations of MNP presence and concentration in tumor, para-tumoral and normal tissues are needed in order to solve these interpretation issues.

## 6. Discussion

In the recent years, the increasing abundance of environmental MNPs and their capability to recirculate, infiltrate and deposit into various ecosystems, including the human body, has raised significant concerns about their long-term health effects. Although the related body of evidence has progressively expanded in recent decades, knowledge about MNP-related health risks, especially as a carcinogen, remains limited to date. Their potential to foster inflammation and oxidative stress in living beings and humans indicates a hypothetical role in cancer initiation and progression. Fragmented and rather initial evidence suggests that MNPs can trigger several biological processes that are strongly linked to carcinogenesis, as tumor-promoting inflammation, oxidative stress and ROS production, direct and indirect genotoxicity (MNPs being carriers of genotoxic chemicals), alterations in immune response and disruption of the gut microbiome (Figure 2). On the other hand, there is an important lack of large-scale epidemiological studies evaluating the impact of human MNP exposure on cancer incidence and mortality. In fact, the IARC has not ever mentioned MNPs as a potential carcinogen for humans, and most specific plastic polymers that have the potential to give origin to secondary MNPs have been listed in group 3, i.e., “not classifiable as to its carcinogenicity to humans”, according to IARC monographs (see https://monographs.iarc.who.int/list-of-classifications).

Current basic research is hampered by the incomplete refinement of analytical methods, and the heterogeneity of MNPs in their sizes, shapes, polymer structure, chemical properties, nature and composition of the eco-corona further complicate experimental result interpretations. At the eco-toxicological level, human exposure potentially depends on geographical location, weather and soil properties, density of MNP emissions, type of diet and daily habits. Reasonably, dissecting the abundance of different MNPs in indoor environments could refine MNP exposure estimation.

In the current state of knowledge, health risks related to MNP exposure, in particular as a potential carcinogen, remain poorly understood. Preclinical evidence shows major biological effects of MNP exposure, but dose expositions are significantly higher in experimental models compared to real-life cellular exposures; thus, results are seldom reliable.

In our view, in order to expand knowledge regarding MNP effects, some major areas of further development should be implemented in the near future:First, standardization in analytical methods for MNP detection in biological specimens will be essential to achieve reproducible and reliable results and to provide the basis for building consistent evidence from different scientific groups.Studies employing mammalian models should evaluate MNPs’ bioaccumulation in tissues over time, thanks to innovative in vivo techniques, as well as unravel the real genotoxic potential of MNPs. In this regard, human organ-specific models could further elucidate tissue-specific effects and provide information on risks specific to different routes of exposure.Translational research should effortlessly characterize the presence, distribution and abundance of MNPs in normal and cancer tissues, aiming at investigating potential prognostic and molecular implications. In the near future, the integration of advanced analytical techniques, computational tools and multi-omics approaches could unravel MNPs’ role as carcinogens.At a more complex level, MNPs could interact with multiple fundamental body-wide entities, such as microbiome, systemic immunity and nervous signals, which have to be further elucidated.To date, there is an urgent need for robust epidemiological data evaluating the correlation of MNPs exposure and disease development at the population level. Longitudinal epidemiologic studies should determine the cancer risk for populations at higher probability of MNP exposure, as in industrial and metropolitan areas or in coastal locations. Moreover, epidemiological data would be extremely valuable in order to generate hypotheses in basic science and further elucidate the underlying mechanisms.

## 7. Conclusions

In recent years, MNPs have raised public health concerns due to their ubiquitous nature as environmental pollutants. Currently, the scarcity of knowledge regarding MNP-related health risks hampers the development of regulatory policies and public health strategies aimed at controlling human MNP exposure.

In the field of oncology, many significant advances have been achieved in recent decades, especially in the characterization of cancer (epi)genic, metabolic, immunological and microenvironmental features. Meanwhile, environmental and behavioral causative factors are rarely identified due to the complex interplay between different risk factors in multifactorial carcinogenesis. Indeed, in recent years, epidemiological data have shown a potential increase in early-onset cancers [110,111,141,142], and this could be paired with potentially increased human MNP exposure in future generations. For these reasons, the scientific community will have to pursue intensive research efforts in order to unravel the real carcinogenic role of MNPs. This knowledge will be fundamental in informing global regulatory ecological and health policies.

## Figures and Tables

**Figure 1 ijms-26-00215-f001:**
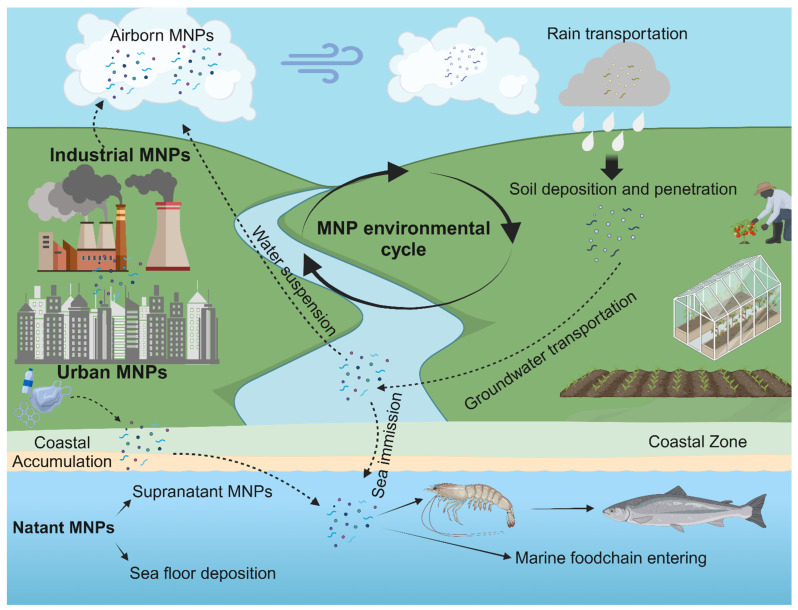
MNP environmental recirculation. Once emitted into the environment, MNPs have the capability to continuously recirculate through air suspension, soil penetration and sea emission, entering the water cycle. Sea natant MNPs can be ingested by marine organisms, persisting in the marine food chain. In addition, airborne MNPs can precipitate within rain, penetrating in soil and reaching underwater, rivers and lakes.

**Figure 2 ijms-26-00215-f002:**
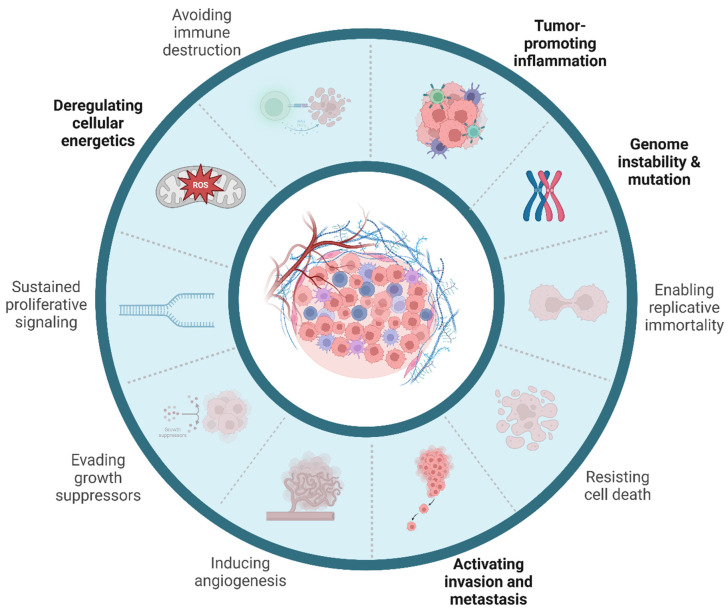
Hallmarks of cancer related to MNPs. Current evidence regarding the role of MNPs in cancer initiation and promotion is linked mainly to the capability of MNPs to induce metabolic stress through the induction of ROS, fostering immune infiltration and chronic inflammation since their persistence in cancer cells and macrophages. The inability of intracellular lytic enzymes of mononuclear phagocytes to process MNPs induces a “frustrated” phenotype that can cause uncontrolled cell death, further sustaining inflammatory processes. Moreover, MNP-exposed cancer cells showe an augmented capacity of invasion and metastatization in preclinical models. Finally, genomic instability could be the result of intricated cytotoxic and genotoxic damage triggered by the presence of MNPs, which needs to be further elucidated. Faded sections indicate that poor or no evidence is available in demonstrating a potential effect in those specific hallmarks.

## Data Availability

Figure 1 created in BioRender. Ruggieri, L. (2024) https://BioRender.com/h88g872. Figure 2 was created in BioRender. Ruggieri, L. (2024) https://BioRender.com/u69b909.

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
