# Peer review of "Rising Concern About the Carcinogenetic Role of Micro-Nanoplastics"

_ijms, 2024, doi:10.3390/ijms26010215_

Round 1
Reviewer 1 Report
Comments and Suggestions for Authors
The manuscript by Ruggieri et al. entitled “Rising concern about the carcinogenetic role of micro-nanoplastics” summarizes the current data on the carcinogenic effects of microplastics, which is a highly relevant topic. The paper is well designed, sections are well selected and located in a suitable order providing the logical information flow. However, in general, the paper lacks in-depth analysis of the molecular mechanisms involved. Multiple sections should be expanded. In particular, exposure routes are indeed described extremely briefly. Exposure routes are well summarized in a recently published review (PMID: 36427812). Mechanisms of microplastics toxicity are covered superficially focusing only on ROS and inflammation. What about different cell death pathways like apoptosis, pyroptosis, ferroptosis, necroptosis, etc.? There is abundant experimental evidence on their role. What about the mitochondrial and lysosomal dysfunction, ER stress, etc.? The toxicity effects should be highlighted in a figure. Which specific signaling pathways are affected by microplastics?
More systems should be considered. What about the effects on the reproductive system, skin, etc. These sections should contain in-depth analysis of molecular signaling involved, which is missing. The same is applied to the sections on carcinogenic effects. In the current state, the manuscript does not provide reliable evidence supporting the carcinogenic effects of microplastics.
Author Response
Reviewer 1
The manuscript by Ruggieri et al. entitled “Rising concern about the carcinogenetic role of micro-nanoplastics” summarizes the current data on the carcinogenic effects of microplastics, which is a highly relevant topic. The paper is well designed, sections are well selected and located in a suitable order providing the logical information flow.
We profoundly thank Reviewer 1 for the appreciation of our manuscript.
However, in general, the paper lacks in-depth analysis of the molecular mechanisms involved. Multiple sections should be expanded. In particular, exposure routes are indeed described extremely briefly. Exposure routes are well summarized in a recently published review (PMID: 36427812).
Since the scope of our review is to summarize only substantial evidence in the field, we avoided to discuss human exposure estimation and excessive discussion about eco-toxicological consideration.
Mechanisms of microplastics toxicity are covered superficially focusing only on ROS and inflammation. What about different cell death pathways like apoptosis, pyroptosis, ferroptosis, necroptosis, etc.? There is abundant experimental evidence on their role. What about the mitochondrial and lysosomal dysfunction, ER stress, etc.? The toxicity effects should be highlighted in a figure. Which specific signaling pathways are affected by microplastics?
We thank Reviewer 1 for these considerations. Although we recognize that abundant evidence is available regarding the induction of pyroptosis, ferroptosis and necroptosis phenomena by microplastics in mouse models, the aim of our review was to discuss general aspects of molecular pathology only. Indeed, we did not describe the specific cellular events that are consequences of ROS production and inflammation, in order to avoid excessive in-depth discussion, which is substantially far beyond the scope of our review. However, we implemented references with articles that analysed the topics cited by Reviewer 1 and added a sentence in the 5.1 section. Finally, lysosomal and mitochondrial dysfunctions were indirectly already cited in the text (i.e. frustrated macrophages and mitochondrial disfunction).
More systems should be considered. What about the effects on the reproductive system, skin, etc. These sections should contain in-depth analysis of molecular signaling involved, which is missing. The same is applied to the sections on carcinogenic effects. In the current state, the manuscript does not provide reliable evidence supporting the carcinogenic effects of microplastics.
We thank Reviewer 1 for the correct consideration. The aim of our Review was primarily to highlight that evidence to date do not support a clear carcinogenic effect of MNPs, but since initial evidence suggest a potential effect, efforts in human research have to dissect this hypothesis in the near future. We aimed primarily in increasing the awareness about the topic, and not in reporting definitive evidence. Finally, we propose different strategy to implement human research from a medical oncologist point-of-view. For the above reasons, we did not provide extensive description of molecular signals involved, but a general overview only.
Reviewer 2 Report
Comments and Suggestions for Authors
The authors of the paper have produced a good, complete, up-to-date and well-structured review. I only have a few comments to make:
1- The reference is missing on line 37.
2- For Figure 1, I think it would be interesting to include the different degrees of sedimentation in aqueous media of the different types of plastics.
3- I miss a reference on line 91, where the prevalence of plastics is correlated with their increase in the environment.
4- When talking about standardized methodologies for the identification of plastics, I think ISO 4484-2:2023 should be mentioned.
5- Is eco-corona a different concept from plasticsphere?
6- I do not think it is necessary to divide section 2.2 into subsections.
7- Has any critical concentration been described at which human exposure to plastics (whether via ingestion, inhalation or dermal absorption) manifests itself in significant metabolic changes? 8- In the current subsection 2.2.2., in which regions are there more chances of inhaling MNPs? Is it known in which regions there are more MNPs in the atmosphere? What does the presence of MNPs in the environment depend on? How do they get there? Please include this information.
9- In the current subsection 2.2.3., have products been described that increase/stimulate the absorption of MNPs?
10- In line 208 I miss a reference after “other physiological processes”.
11- The role of plastics as hormone disruptors should be discussed, even if briefly.
12- Justify the paragraphs between lines 347 and 370.
Author Response
Reviewer 2
The authors of the paper have produced a good, complete, up-to-date and well-structured review. I only have a few comments to make:
- The reference is missing on line 37.
We really apologize for this missing. We added the correct reference.
- For Figure 1, I think it would be interesting to include the different degrees of sedimentation in aqueous media of the different types of plastics.
Although we acknowledge the importance of different degrees of sedimentation in freshwater and seas in relation to types and dimension of micro-nanoplastics, as highlighted by Reviewer 2, an excessive level of insight into eco-toxicological discussion is beyond the scope of our review.
- I miss a reference on line 91, where the prevalence of plastics is correlated with their increase in the environment.
As suggested by Reviewer 2, we added a reference in this regard.
- When talking about standardized methodologies for the identification of plastics, I think ISO 4484-2:2023 should be mentioned.
We are grateful to Reviewer 2 for this important suggestion. We mentioned ISO in the text as indicated.
- Is eco-corona a different concept from plasticsphere?
Not properly. Indeed, eco-corona is a microscopic type of plastisphere, since the latter indicates also macroscopic ecological environments that develop on the surface of macroplastics. Actually, eco-corona, the biophysical plastisphere around microplastics, is probably more common. In conclusion, eco-corona is a well-recognized and particular entity related to microplastics only. We profoundly thank Reviewer 2 for the inspirational discussion.
- I do not think it is necessary to divide section 2.2 into subsections.
We modified the manuscript according to Reviewer 2 comment and we are grateful for this substantial improvement in readability.
- Has any critical concentration been described at which human exposure to plastics (whether via ingestion, inhalation or dermal absorption) manifests itself in significant metabolic changes? In the current subsection 2.2.2., in which regions are there more chances of inhaling MNPs? Is it known in which regions there are more MNPs in the atmosphere? What does the presence of MNPs in the environment depend on? How do they get there? Please include this information.
We thank Reviewer 2 for the considerations. To date, no informations are available regarding the potential critical concentration at which human exposure convert into metabolic changes. As we discussed above in the 2.1 section, where we provide general eco-toxicological considerations, since a substantial amount of MNPs is emitted in industrial zones and cities, human exposure is thought to be higher in there, although no data on human exposure are available to date. Mismanaged environmental plastics accumulate in coastal zones and recirculate in marine and sea ecosystems, reaching atmosphere (as showed in Figure 1). Indeed, MNPs are reported to be much concentrated in both land and marine coastal zones, but data from determined areas are poor (as discussed in section 2.1). However, detailed discussion about this topic is beyod the scope of our review.
- In the current subsection 2.2.3., have products been described that increase/stimulate the absorption of MNPs?
To date, identification of absorption enhancers is outstandingly challenging since evidence regarding the absorption of micro-nanoplastic are still poor. In general, we can say that eco-corona is the primary enhancer of micro-nanoplastic penetration in living beings and humans, since its biological components can facilitate the interaction between cell surface receptors and mucous layers, allowing the transition of micro-nanoplastics in tissues. We are grateful to Reviewer 2 for the interesting discussion.
- In line 208 I miss a reference after “other physiological processes”.
Since that sentence is a mere introduction to the discussion below, we did not provide a specific reference. The other physiological processes are described immediately after in the paragraph.
- The role of plastics as hormone disruptors should be discussed, even if briefly.
We thank Reviewer 2 for the important suggestion. We briefly discussed the topic in the section 4.4. We apologize for this missing in the previous version.
- Justify the paragraphs between lines 347 and 370.
We are sorry for this error. We justified the text part as Reviewer 2 suggested.
Reviewer 3 Report
Comments and Suggestions for Authors
The authors present an interesting review which attempts to capture the current understanding of micro and nanoplastics on human physiology; in particular, cancer. Briefly, the authors give a good account of the source of these plastic particulates, and the routes through which they enter the human body. Thereafter, a summary of the established effects is given while models for investigating and methods to detect the plastic materials are also covered. Together, this review captures much of what is recognised/purported with small plastic matter and human health in a broad and engaging way.
In reviewing the manuscript I had a number of concerns. The authors should address the following when preparing a suitable revision.
1. Can the authors confirm the definition of a ‘micro nanoplastic’? I have encountered microplastics and nanoplastics, and I am wondering if the authors intend to capture both in using MNP, or is a micro nanoplastic distinct in its own right? This is somewhat elaborated upon, but I think clearer definition of what is intended with this abbreviation is needed.
2. There are instances in the review where [ref] appears where a reference number should appear. The authors should revise the manuscript and address these instances as appropriate.
3. I think more information on the chemical nature of the MNPs is needed in order to discern the differences that can be encountered in their effects, the eco-corona they develop, etc.
4. Some sections are weak on referencing in general whereby some paragraphs/long blocks of text rely on a single reference or two. In general, the review needs improvement on the amount of evidence being used to support certain topics.
5. Moreover, some sections take a particularly broad view of their topic, and some sections would benefit from discussing some mechanisms/events in more detail. For example, many parts of 2.2 ‘Routes of Human Exposure’ are broad and would benefit having more instances of references/studies which demonstrated proof that these MNPs interact with these structures. This is just one example, but more sections suffer a similar effect.
6. Some information in the text is quite repetitive, and some points are made several times over. The authors should review the writing and improve on this.
7. The authors should consider including a table to capture some of the effects highlighted within. This might be an economical way of presenting several studies pertaining to each physiological system for example rather than add to the existing text.
8. Overall, the writing is interpretable for the most part, but there are instances of typos throughout. For example, in section 2.2.2 and 2.2.3 the paragraphs start off with lower case letters, and section 3.3’s title has all capitals for the first word i.e. EX. These are minor, but noticeable and all instances should be addressed in advance of any resubmission.
9. In Figure 2, some of the icons are quite faded, while some are ‘bolder’. Is this intentional? Does whether the icons are faded/bolder signify something?
Author Response
Reviewer 3
The authors present an interesting review which attempts to capture the current understanding of micro and nanoplastics on human physiology; in particular, cancer. Briefly, the authors give a good account of the source of these plastic particulates, and the routes through which they enter the human body. Thereafter, a summary of the established effects is given while models for investigating and methods to detect the plastic materials are also covered. Together, this review captures much of what is recognised/purported with small plastic matter and human health in a broad and engaging way.
In reviewing the manuscript I had a number of concerns. The authors should address the following when preparing a suitable revision.
- Can the authors confirm the definition of a ‘micro nanoplastic’? I have encountered microplastics and nanoplastics, and I am wondering if the authors intend to capture both in using MNP, or is a micro nanoplastic distinct in its own right? This is somewhat elaborated upon, but I think clearer definition of what is intended with this abbreviation is needed.
We defined MNPs in the introduction section. We reported the difference in size between microplastic (MP) and nanoplastic (NP) and we mentioned that MP and NP together form the environmental MNP debris. In the current literature, MNPs are named micro/nanoplastics or micro(nano)plastics, since they represent the two part of the same entity (micro-nanoplastic debris o microplastic debris/ litter) considering the cutoff of 1 micrometer (MP > 1 micrometer, NP < 1 micrometer). Nonetheless, there are no standardized cutoffs to date. We considered this cutoff as the best suitable for biological considerations. We thank Reviewer 3 for the consideration.
- There are instances in the review where [ref] appears where a reference number should appear. The authors should revise the manuscript and address these instances as appropriate.
We really apologize for this missing. We added the correct reference.
- I think more information on the chemical nature of the MNPs is needed in order to discern the differences that can be encountered in their effects, the eco-corona they develop, etc.
We are profoundly grateful to Reviewer 3 for this important suggestion. We expanded the discussion regarding this fundamental topic in the dedicated section (paragraph 2.1). However, excessive in-depth discussion about chemical and physical properties of different polymers in relation to the different eco-corona they can develop is beyond the scope of our review.
- Some sections are weak on referencing in general whereby some paragraphs/long blocks of text rely on a single reference or two. In general, the review needs improvement on the amount of evidence being used to support certain topics.
We thank Reviewer 3 for the suggestion that has consistently improved our manuscript. We balanced references throughout the text to support our consideration with the available evidence.
- Moreover, some sections take a particularly broad view of their topic, and some sections would benefit from discussing some mechanisms/events in more detail. For example, many parts of 2.2 ‘Routes of Human Exposure’ are broad and would benefit having more instances of references/studies which demonstrated proof that these MNPs interact with these structures. This is just one example, but more sections suffer a similar effect.
We thank Reviewer 3 for the correct suggestion. Since evidence regarding the human exposure are fragmented and often are represented by mere estimations, we did not add a level of detail in order to excessively discuss topics without substantial evidence. Since the scope of our review is to summarize only the evidence with adequate quality available to date, some sections are inevitably short or general.
- Some information in the text is quite repetitive, and some points are made several times over. The authors should review the writing and improve on this.
As suggested by Reviewer 3, we tried to improve writing and discussion throughout the text, eliminating excessive repetitions.
- The authors should consider including a table to capture some of the effects highlighted within. This might be an economical way of presenting several studies pertaining to each physiological system for example rather than add to the existing text.
We added a table (Table 1) as suggested by Reviewer 3. We are grateful for the precious suggestion that improve the readability of the manuscript.
- Overall, the writing is interpretable for the most part, but there are instances of typos throughout. For example, in section 2.2.2 and 2.2.3 the paragraphs start off with lower case letters, and section 3.3’s title has all capitals for the first word i.e. EX. These are minor, but noticeable and all instances should be addressed in advance of any resubmission.
We addressed the typos that Reviewer 3 suggested. We apologize for this inconvenience.
- In Figure 2, some of the icons are quite faded, while some are ‘bolder’. Is this intentional? Does whether the icons are faded/bolder signify something?
Thank to Reviewer 3 comment, we realized that Figure 2 was not really understandable. We improved the Figure 2 caption to highlight that faded sections means that overall evidence in those fields is still lacking.
Round 2
Reviewer 1 Report
Comments and Suggestions for Authors
The authors have provided the explanation why the comments have not been addressed.
Author Response
We are profoundly grateful to Reviewer 1 for understanding our paper's aim. We hope that the message we want to provide will easily reach readers.
Moreover, we improved the manuscript with additional references.
Our best regards.
Reviewer 3 Report
Comments and Suggestions for Authors
The authors have responded to my comments, and while some of the changes have improved the manuscript, there are aspects that remain a concern. Given the feedback provided I expected a greater level of change to have occurred with the writing and as such some of my points still stand.
Author Response
We thank Reviewer 3 for the previous important comments. We deeply reviewed our manuscript for possible excessive repetitions, but we did not find particular concerns in the current version. In addition, we further clarified the aim of our paper, to facilitate the reader's focus.
For the abovementioned reasons, we added only minor modifications to the text.
We thank Reviewer 3 in advance for her/his comprehension.